# FoundationMotion: Auto-labeling and Reasoning about Spatial Movement in Videos

## Abstract

Motion understanding is fundamental to physical reasoning, enabling models to infer dynamics and predict future states. However, state-of-the-art models still struggle on recent motion benchmarks, primarily due to the scarcity of large-scale, fine-grained motion datasets. Current approaches rely heavily on costly manual annotation, severely limiting scalability. To address this challenge, we introduce FoundationMotion, a fully automated data curation pipeline that constructs large-scale motion datasets. Our approach first detects and tracks objects in videos to extract their trajectories, then leverages these trajectories and video frames with large language models to generate fine-grained captions and diverse question–answer pairs about motion and spatial reasoning. Using datasets produced by this pipeline, we fine-tune open-source models including NVILA-Video-15B and Qwen2.5-7B, achieving substantial improvements in motion understanding without compromising performance on other tasks. Notably, our models outperform strong closed-source baselines like Gemini-2.5 Flash and large open-source models such as Qwen2.5-VL-72B across diverse motion understanding datasets and benchmarks. FoundationMotion thus provides a scalable solution for curating fine-grained motion datasets that enable effective fine-tuning of diverse models to enhance motion and spatial reasoning capabilities.

## 1 Introduction

> *"Spatial thinking is the foundation of thought."*
>
> — Barbara Tversky, *Mind in Motion: How Action Shapes Thought*

In *Mind in Motion* (Tversky, 2019), psychologist Barbara Tversky argues that spatial cognition is not a secondary aspect of thought but a foundational one. It enables us to make sense of the world through our physical actions and interactions. These real-world movements become internalized as mental operations, often expressed spontaneously through gestures. Moreover, spatial thinking supports a wide range of everyday and expert activities, from using maps and assembling furniture to designing systems and understanding flows of people, traffic, or information. Whether estimating how to parallel park, imagining how to fold a piece of paper into a shape, mentally rotating an object, or figuring out how to carry multiple items through a narrow doorway, we rely on a powerful yet often overlooked capacity: spatial thinking. Motivated by this insight, our goal is to enable machines to effectively describe and reason about object motion, allowing them to understand and reason in the physical world as humans do through the development of robust Vision-Language Models (VLMs). To ground this effort, we focus on learning from videos, where motion and spatial interactions unfold over time.

Reflecting on the rapid advancement of VLMs, significant progress has been made in learning from videos (Liu et al., 2025; Weng et al., 2024; Chen et al., 2024; 2025). State-of-the-art models such as Gemini (Comanici et al., 2025) and Qwen-VL (Bai et al., 2025; Wang et al., 2024) demonstrate impressive capabilities in identifying objects and interpreting complex environments and events. However, despite these advances, current VLMs still face considerable challenges in fully understanding the nuanced spatial and motion dynamics inherent in many real-world videos. Addressing these challenges is crucial for enabling machines to reason about the physical world as effectively as humans do. For instance, while Gemini models achieve remarkable results in understanding objects,

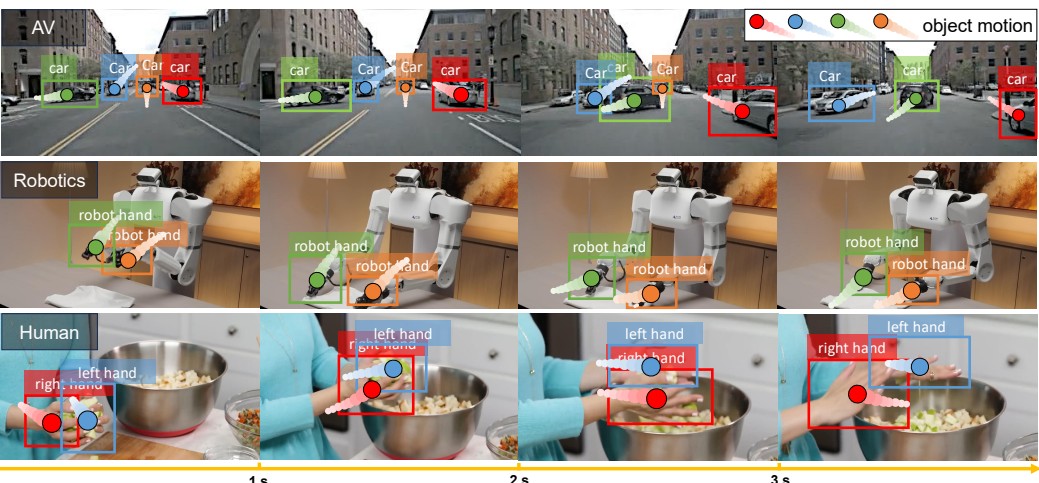

Figure 1: **Illustration of motion automatically labeled using FoundationMotion**. Our proposed FoundationMotion automatically detects and tracks moving objects, annotating their spatial movement (motion) in videos. We demonstrate the auto-labeled motion trajectories on diverse video domains, including autonomous driving, robotics, and human daily activities.

scenes, and events in videos, they sometimes fail to recognize basic object motion, such as "the car is turning right," which is a relatively simple task for humans. These limitations pose serious threats when deploying these foundation models in real-world embodied applications such as robotics and autonomous driving. This is because we need machines to understand not only "what is this motion" (e.g., pouring water) but also "how this motion happens" (e.g., pouring water from a bottle into a glass). Recent state-of-the-art methods such as PerceptionLM (Cho et al., 2025) and NVILA (Liu et al., 2025) have excelled at understanding "what" but still face challenges in understanding "how." We attribute this primarily to the lack of "how" motion data.

However, creating "how" motion data is quite challenging. Building a robust VLM that can generalize in understanding spatial movement and object motion requires accurate training data in object detection, tracking, and linking behaviors to specific motions. This means an annotator might need several minutes to label just a 3-second video. It would take a team of 10 people approximately 100 days to complete annotations for 100,000 videos. When videos may vary in length from a few seconds to several minutes or even hours, the cost and time required increase significantly, not to mention the challenge of ensuring annotation quality. To address this challenge, we propose **FoundationMotion**, a fully automated and unified data curation pipeline for large-scale object motion understanding. FoundationMotion leverages state-of-the-art recognition models (e.g., Segment Anything V2) and LLM-based reasoning to detect, track, and annotate object motion across diverse videos (see Figure 1 for examples of our auto-labeled visualizations). It focuses on motion-centric video cropping, object detection (e.g., vehicles, hands, bodies), and multi-object tracking, generating structured motion data. These annotations are then aggregated and distilled into descriptive motion summaries using Large Language Models (LLMs), enabling both motion understanding and question-answering over dynamic scenes.

In summary, our main contributions are as follows:

1. We propose **FoundationMotion**, a fully automated, unified data curation pipeline that constructs large-scale motion datasets for accurate detection, tracking, and understanding of object behavior. Based on this auto-labeling pipeline, we generate approximately 500K question-answering pairs (QAs) and captions, collectively referred to as the **FoundationMotion Dataset**.

2. To address the lack of "how" motion benchmarks, we manually collect videos of varying lengths and annotate QAs across multiple domains, including hand motion in human daily activities and driving, robot motion during manipulation tasks, and car motion in autonomous driving.

3. We fine-tune several open-source VLMs with our FoundationMotion Dataset and evaluate the results on both public, widely-used benchmark (primarily focusing on "what" behavior) and our manually annotated "how" motion benchmark . Our results demonstrate that models fine-tuned on the FoundationMotion dataset achieve superior performance compared to larger open-source models and even closed-source models such as Gemini-2.5-Flash.

4. We will release all our code, data, and benchmarks. We hope that FoundationMotion will raise awareness about the importance of motion understanding, establish a standard for the field, and foster community development. Continuous efforts and improvements will be made to refine the FoundationMotion codebase and dataset.

## 2 RELATED WORK

### 2.1 MOTION-FOCUSED VIDEO UNDERSTANDING BENCHMARKS

Recent work has introduced benchmarks for fine-grained motion understanding in videos. MotionBench (Hong et al., 2025) evaluates basic motion-level perception through granular movement questions, revealing that state-of-the-art video VLMs score below 60%, highlighting a significant deficiency in motion reasoning. FAVOR-Bench (Tu et al., 2025) further expands this evaluation with 1,776 curated videos and thousands of Q&A pairs across categories such as sequential actions and camera motions, alongside a training set (FAVOR-Train). However, evaluations across 21 multimodal LLMs demonstrated performance far below human level.

MotionBench and FAVOR-Bench emphasize fine-grained motion recognition (what moves, when, and how detailed) but overlook spatial reasoning (how motions interact, relative trajectories, geometric constraints). We fill these gaps by enabling models to capture spatial relations and by addressing data scarcity: instead of relying on limited or manually curated data, we construct a large-scale dataset with a fully automated pipeline. Training on it produces foundation models with state-of-the-art motion reasoning, serving as both a benchmark and training resource for advancing fine-grained motion understanding.

### 2.2 AUTOMATED VIDEO DATASET CONSTRUCTION AND ANNOTATION

Manual video annotation for captioning or QA is costly, so recent work has shifted to automated pipelines. VideoEspresso (Han et al., 2025) used LLMs to generate a large-scale VideoQA dataset, scaling beyond crowdsourcing. CinePile (Rawal et al., 2024) produced 305k QA pairs for long movies via LLM prompting with audio descriptions, enabling complex temporal and narrative queries. VoCap (Uijlings et al., 2025) auto-captioned objects using segmentation masks and vision-language models, improving object-centric captioning. UltraVideo (Xue et al., 2025) applied motion-based filters to retain only informative clips.

Our data generation pipeline extends this paradigm with a focus on fine-grained object motions. Unlike prior work, it applies multi-object tracking and automatically generates detailed captions and QA pairs about object trajectories. This yields a dataset tailored to spatial object behavior, filling the gap left by earlier QA- or captioning-focused efforts and enabling models to acquire motion-centric knowledge at a scale and granularity that would be infeasible with manual labeling.

### 2.3 VISION-LANGUAGE VIDEO FOUNDATION MODELS

Recent advances in vision-language video models extend LLMs to video understanding, enabling captioning, QA, and retrieval, yet they struggle with fine-grained motion and spatio-temporal reasoning. MotionBench (Hong et al., 2025) shows that leading models (e.g., InternVideo (Wang et al., 2022), Video-LLaMA (Zhang et al., 2023)) remain weak in motion understanding. Meanwhile, PerceptionLM (Cho et al., 2025) stresses perceptual grounding with open-access data, and Locate3D (Arnaud et al., 2025) improves object-centric spatial reasoning via self-supervised 3D localization but still fails to capture how motion happens.

We address this gap by introducing a motion-aware vision-language model explicitly trained with our new fine-grained motion dataset. Infusing such data enables strong performance on motion recognition, localization, and reasoning while preserving broad video-language capabilities. Unlike

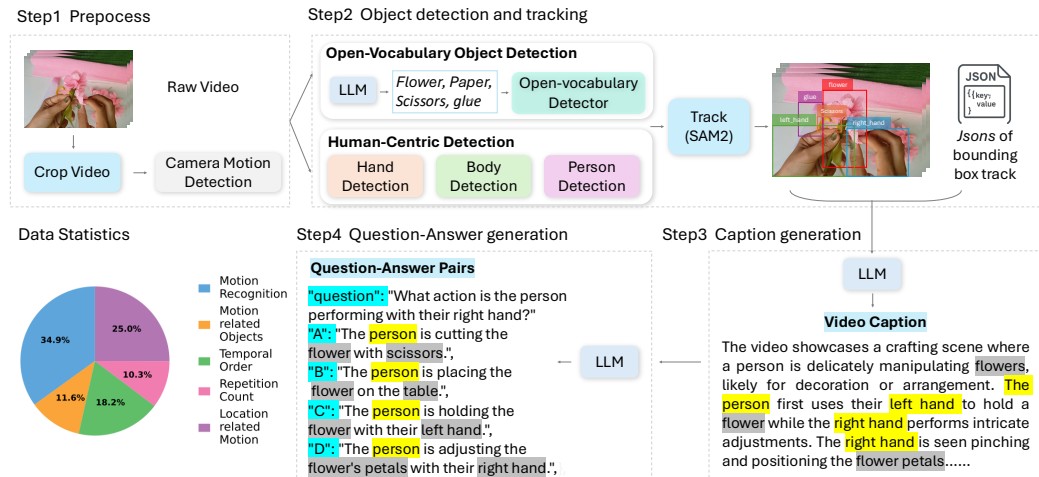

Figure 2: **FoundationMotion Data Curation Pipeline**. FoundationMotion is a fully automated pipeline for constructing large-scale motion datasets, enabling accurate detection, tracking, and understanding of object behavior. It leverages recognition models (e.g., Segment Anything) and understanding models (e.g., LLMs). Videos are first cropped to focus on motion, then objects such as cars and human-centric items (hands, bodies, persons) are detected and tracked. Their location changes are annotated into JSON files, which are summarized into captions. Finally, we design specific prompts for the LLM to generate questions and answers.

prior models that lacked targeted motion training, our approach demonstrates that motion-focused learning can both improve motion understanding and enhance overall versatility.

## 3 FOUNDATIONMOTION DATA CURATION PIPELINE

**Overview.** Training a high-capability video motion model requires large-scale data, yet manually annotating fine-grained motion in videos is costly and time-consuming. This motivates the need for an automated data curation pipeline. While LLMs have shown remarkable progress in building automated pipelines across several domains, their ability is constrained when given only raw video input: they can handle simple object and action recognition but struggle to capture spatial relations and complex motions. In parallel, recent advances in vision models have demonstrated strong capabilities in detection, tracking, and summarization. Building on these complementary strengths, we design a fully automated data curation pipeline that produces detailed motion annotations and corresponding question–answer (QA) pairs from videos, as illustrated in Figure 2. In the following, we describe its four stages in detail: video preprocessing (Sec. 3.1), object detection and tracking (Sec. 3.2), caption generation (Sec. 3.3), and QA generation (Sec. 3.4).

### 3.1 VIDEO PREPROCESSING

The preprocessing stage prepares raw videos for downstream analysis by performing temporal cropping and frame extraction. Given an input video $V$ with duration $t_v$, we extract a temporal segment of 5–10 seconds. If $t_v \leq 5$ seconds, the entire video is retained. For longer videos, we sample a segment with duration $t_s \sim \mathcal{U}(5, \min(10, t_v))$, centered approximately at the midpoint of the video:

$$t_{\text{start}} = \max\left(0, \min(t_v - t_s, t_{\text{mid}} + \epsilon)\right),$$

where $t_{\text{mid}} = \frac{t_v}{2} - \frac{t_s}{2}$ denotes the centered position and $\epsilon \sim \mathcal{U}(-0.2t_v, 0.2t_v)$ introduces temporal variation. This strategy yields representative segments while controlling computational costs.

When the camera moves together with the object, even humans find it difficult to describe the object's motion. To ensure the model can learn clear spatial relations, we employ VGGT (Wang et al., 2025) to detect and filter videos with significant camera motion. The model predicts camera poses across sampled frames, computing motion scores based on translation and rotation changes between

consecutive frames. We compute the motion score as $s_m = \alpha \cdot \Delta_t + \beta \cdot \Delta_r + \gamma \cdot \max(\Delta_t) + \delta \cdot \max(\Delta_r)$, where $\Delta_t$ and $\Delta_r$ represent average translation and rotation changes, respectively. Videos exceeding a motion threshold $\tau_{motion} = 0.3$ are excluded from further processing, as camera motion significantly degrades tracking quality and annotation accuracy.

### 3.2 OBJECT DETECTION AND TRACKING

Our object detection is divided into two components: open-vocabulary object detection (Sec 3.2.1) and human-centric detection (Sec 3.2.2). We first design an open-vocabulary detection pipeline to identify all general objects in the images. we also introduce a tailored human-centric detector specialized for detecting humans, left hands, right hands, and objects held in hands, since distinguishing between the left and right hands is particularly challenging for standard detectors.

#### 3.2.1 OPEN-VOCABULARY OBJECT DETECTION

We leverage the Qwen2.5-VL-7B model (Bai et al., 2025) to analyze the first frame and identify salient objects within the scene. Specifically, the model produces a set of object categories $\mathcal{O} = \{o_1, o_2, \ldots, o_n\}$ in the video via natural language generation, providing high-level semantic coverage of the scene content. Given these object categories, we employ Grounded-DINO (Liu et al., 2023) to localize objects precisely, yielding $\mathcal{B}_{obj} = \text{GroundedDINO}(I_0, \mathcal{O})$, where $I_0$ denotes the first frame and $\mathcal{B}_{obj}$ corresponds to the detected bounding boxes with class labels. We query Grounded-DINO with individual object classes rather than concatenating all classes into a single prompt. This enforces a one-to-one alignment between detected boxes and semantic labels, thereby improving detection quality.

#### 3.2.2 HUMAN-CENTRIC DETECTION

For human motion understanding, we adopt a hierarchical pipeline that refines detection from person- to hand-level. Person detection uses Cascade Mask R-CNN with a ViTDet-H backbone (Li et al., 2022), ensuring robust localization with high confidence ($\tau_{person} = 0.8$). Each detected person is then processed by ViTPose+ (Xu et al., 2022) to extract whole-body keypoints, including 42 hand keypoints that define initial hand regions, later expanded by 1.5× to cover pose variations. The Hands23 model (Cheng et al., 2023) performs hand detection with contact-state and hand–object interaction analysis. For each hand $h_i$, it predicts $(b_h^i, s_h^i, c_h^i, b_o^i)$, where $b_h^i \in \mathbb{R}^4$ is the bounding box, $s_h^i \in \{\text{left}, \text{right}\}$ the laterality, $c_h^i \in \{\text{no\_contact}, \text{self\_contact}, \text{object\_contact}, \text{other\_contact}\}$ the contact state, and $b_o^i$ the object box if $c_h^i = \text{object\_contact}$. Hand–person associations are established via IoU matching between ViTPose regions and Hands23 detections, requiring IoU $> 0.3$.

#### 3.2.3 TEMPORAL TRACKING

Temporal coherence is maintained through SAM2 (Ravi et al., 2024), which propagates detections across video frames using a two-stage tracking strategy. In the initial tracking stage, person and object bounding boxes from the first frame initialize SAM2's video predictor. Each entity receives a unique identifier following a hierarchical scheme: persons are assigned IDs in the range $[0, 99]$ with sub-IDs for associated body parts (ID $\times 10$ for person, ID $\times 10 + 1$ for left hand, ID $\times 10 + 4$ for right hand), while objects receive IDs starting from 1000. This ID allocation enables consistent tracking while maintaining semantic relationships between entities.

The refined tracking stage incorporates hand and hand-object detections at keyframes (every 5th frame) to maintain tracking accuracy throughout the video. The propagation follows: $\mathcal{M}_t = \texttt{SAM2.propagate}(\mathcal{M}_{t-1}, \mathcal{B}_{new})$, where $\mathcal{M}_t$ represents segmentation masks at frame $t$ and $\mathcal{B}_{new}$ contains newly detected bounding boxes. This iterative refinement prevents tracking drift while maintaining temporal consistency across extended sequences.

### 3.3 CAPTION GENERATION

The caption generation module uses GPT-4o-mini (Hurst et al., 2024) to transform tracking outputs into natural language. Inputs to GPT-4o-mini include (i) video frames sampled at 2 fps, (ii) structured motion data in JSON containing normalized bounding box trajectories, and (iii) visual

Figure 3: Examples of four zero-shot FoundationMotion evaluation benchmark.

overlays with color-coded bounding boxes. The structured data encodes explicit spatial and temporal information, enabling fine-grained cross-frame reasoning. Caption generation is guided by a prompt covering seven dimensions of motion: (1) action and gesture recognition, (2) temporal sequencing, (3) object–action associations, (4) spatial context, (5) repetition patterns, (6) motion dynamics (direction, distance, velocity, trajectory), and (7) evolving spatial relationships. This structured prompting yields comprehensive and consistent captions capturing both fine-grained motion and high-level semantics.

## 3.4 QUESTION-ANSWER GENERATION

The QA generation stage creates evaluation questions from captions to assess motion and spatial understanding. GPT-4o-mini is prompted with both captions and video frames to produce multi-choice questions targeting specific skills within a structured framework. We design five categories: (1) motion recognition, identifying entity actions; (2) temporal ordering, capturing event sequences; (3) action–object association, linking actors and actions; (4) location-based motion, grounding actions spatially; and (5) repetition counting, recognizing action frequency and patterns. Each question has four options, with distractors drawn from video content, and correct answers randomly distributed to avoid position bias.

## 4 FINE-TUNING WITH FOUNDATIONMOTION FOR STATE-OF-THE-ART MOTION UNDERSTANDING

### 4.1 EXPERIMENTAL SETUP

**Training data.** For training, we take videos from InternVid (Wang et al., 2023), randomly extract 5-second clips from each video, and use the proposed auto-labeling pipeline to obtain captioning and QA data for each video clip. This results in a total of 467K caption/QA-video pairs.

**Evaluation data.** We evaluate our model on both public benchmarks and self-labeled benchmarks. The public benchmarks include MotionBench (Hong et al., 2025) and VLM4D (Zhou et al., 2025), two common benchmarks that evaluate motion understanding in videos. Concretely, MotionBench is a benchmark for fine-grained motion understanding covering six motion tasks, built from internet videos, public datasets, and Unity-simulated data, and containing 5,385 videos with 8,052 carefully human-annotated QA pairs. VLM4D is a benchmark that is specifically designed to test the spatiotemporal reasoning capability of VLMs and contains 1800 QA pairs over 1000 videos that are either from real world or simulated.

The self-labeled benchmarks ("how" motion benchmark), on the other hand, are curated to test the model's zero-shot generalizability to out-of-distribution videos. Specifically, we evaluate motion understanding in daily scenes, autonomous vehicles (AV) and robotic scenarios, which are different from the training videos. For daily scenes, we source videos from 100 Days of Hands (Shan et al., 2020) and manually label 832 QA pairs that are focused on hand motions and hand-object interactions, we refer to this benchmark as **Daily**. Similarly, we collect robotic videos from YouTube and manually label 102 QA pairs on robot motions (**Robotics**), primarily on the robot's hands. We also collect videos from widely used Nuscenes dataset (Caesar et al., 2020) and turn the official manually annotated motion captions (Li et al., 2025) into 1,968 QA pairs that focus on cars' motion (**AV-Car**)

Table 1: Comparison on motion benchmarks. Accuracy gains/losses are marked green/red. The highest and second highest value marked with **bold** / underline. Results are percentages (%). Our **FoundationMotion** dataset consistently boosts performance across benchmarks and yields larger gains than PLM when fine-tuned with the same number of examples. Training with FoundationMotion data brings signiciant improvement on various motion tasks.

| Model | MotionBench | VLM4D | AV-Car | AV-Hand | Daily | Robotics |
|---|---|---|---|---|---|---|
| Gemini-2.5-Flash | 55.6 | **54.7** | 84.1 | **72.7** | 75.4 | 36.1 |
| Qwen-2.5-VL-72B | **61.4** | 50.5 | 83.3 | 56.5 | 80.2 | 36.7 |
| NVILA-Video-15B | 45.7 | 51.8 | 84.4 | 58.1 | 76.2 | 21.4 |
|    FT w/ FoundationMotion | 46.7$_{+1.0↑}$ | 51.9$_{+0.1↑}$ | **91.5**$_{+7.1↑}$ | 58.7$_{+0.6↑}$ | 78.6$_{+2.4↑}$ | 36.3$_{+14.9↑}$ |
|    FT w/ PLM 467k | 47.5$_{+1.8↑}$ | 52.9$_{+1.1↑}$ | 79.4$_{-5.0↓}$ | 55.6$_{-2.5↓}$ | 77.1$_{+0.9↑}$ | 27.4$_{+6.0↑}$ |
| NVILA-Video-8B | 42.3 | 49.0 | 88.9 | 54.6 | 79.1 | 20.4 |
|    FT w/ FoundationMotion | 42.9$_{+0.6↑}$ | 52.4$_{+3.4↑}$ | 90.6$_{+1.7↑}$ | 61.4$_{+6.8↑}$ | **81.1**$_{+2.0↑}$ | **38.2**$_{+17.8↑}$ |
|    FT w/ PLM 467k | 43.6$_{+1.3↑}$ | 49.1$_{+0.1↑}$ | 87.9$_{-1.0↓}$ | 56.0$_{+1.4↑}$ | 75.0$_{-4.1↓}$ | 26.5$_{+6.1↑}$ |
| Qwen-2.5-VL-7B | 39.1 | 41.7 | 80.3 | 47.2 | 61.4 | 28.3 |
|    FT w/ FoundationMotion | 41.3$_{+2.1↑}$ | 44.9$_{+3.2↑}$ | 82.1$_{+1.8↑}$ | 52.8$_{+5.6↑}$ | 73.1$_{+11.7↑}$ | 32.5$_{+4.2↑}$ |

and 108 QA pairs that focus on hands' motion (**AV-Hand**). Therefore, we establish four zero-shot motion benchmarks: AV-car, AV-hand, Daily, and Robotics, with examples from each benchmark shown in Figure 3. We emphasize that there is no overlap between the FoundationMotion dataset and the evaluation benchmarks, which means the results are fully zero-shot.

**Baselines.** We compare our models with state-of-the-art open- and closed-source VLMs including *Gemini-2.5-Flash* (Comanici et al., 2025), *Qwen-2.5-VL-72B/7B* (Bai et al., 2025), and *NVILA-Video-15B/8B* (Liu et al., 2025). To evaluate the quality of our dataset, we compare with PLM (Cho et al., 2025) dataset, a large-scale motion-targeted video QA dataset, by fine-tuning the same model on either our data or PLM data and compare the performances. For fair comparison, we randomly sample 467K instances from PLM dataset such that it has the same amount of data as ours.

**Implementation Details.** Our experiments are conducted on A100 GPUs. For Qwen-related training, we use llamafactory (Zheng et al., 2024) and follow the recommended settings with a learning rate of $10^{-5}$. For NVILA-related training, we follow the official settings (Liu et al., 2025) and set the learning rate to $1.5 \times 10^{-5}$. We apply a cosine annealing schedule and choose Adam as the optimizer. No weight decay is applied.

### 4.2 MAIN RESULTS

**Using FoundationMotion data for fine-tuning yields clear gains across benchmarks and datasets.** With *NVILA-Video-15B*, FoundationMotion lifts MotionBench by *+1.0%*, AV-Car by *+7.1%*, and Robotics by *+14.9%*, while also providing smaller but consistent gains on VLM4D *(+0.1%)*, AV-Hand *(+0.6%)*, and Daily *(+2.4%)*. For *NVILA-Video-8B*, FoundationMotion data improves MotionBench by *+0.6%*, AV-Car by *+6.8%*, and Robotics by *+17.8%*. Similarly, for *Qwen-2.5-VL-7B*, FoundationMotion delivers broad gains across MotionBench *(+2.1%)*, VLM4D *(+3.2%)*, AV-Car *(+1.8%)*, AV-Hand *(+5.6%)*, Daily *(+11.7%)*, and Robotics *(+4.2%)*. These results demonstrate consistent improvements across diverse motion and spatial reasoning tasks.

**Compared with PLM data, fine-tuning on our data with the same budget gives bigger improvements and avoids performance drops.** Compared with PLM, our dataset yields larger and more consistent gains with the same number of examples. On *NVILA-Video-15B* (FoundationMotion vs PLM ), FoundationMotion surpasses PLM on AV-Car *(+7.1% vs. -5.0%)*, AV-Hand *(+0.6% vs. -2.5%)*, Daily *(+2.4% vs. +0.9%)*, and Robotics *(+14.9% vs. +6.0%)*, with PLM slightly better only on MotionBench *(+1.0% vs. +1.8%)* and VLM4D *(+0.1% vs. +1.1%)*. On *NVILA-Video-8B*, our dataset again dominates: VLM4D *(+3.4% vs. +0.1%)*, AV-Car *(+1.7% vs. -1.0%)*, AV-Hand *(+6.8% vs. +1.4%)*, Daily *(+2.0% vs. -4.1%)*, and Robotics *(+17.8% vs. +6.1%)*, while slightly underperforming on MotionBench *(+0.6% vs. +1.3%)*. These results demonstrate that the FoundationMotion dataset provides higher-quality supervision than an equal amount of PLM data.

**With FoundationMotion data, 15B and 7B models surpass Gemini-2.5-Flash and Qwen-2.5-VL-72B on several motion tasks.** FoundationMotion-tuned models can even outperform much larger models like *Gemini-2.5-Flash* and *Qwen-2.5-VL-72B* on several tasks. With *NVILA-Video-15B + FoundationMotion*, AV-Car reaches *91.5%*, surpassing *Gemini-2.5-Flash (84.1%)* and *Qwen-2.5-VL-72B (83.3%)*. The same model also exceeds *Qwen-72B* on VLM4D *(51.9% vs. 50.5%)* and AV-Hand *(58.7% vs. 56.5%)*. These results show that mid-sized open models, when fine-tuned with FoundationMotion, can surpass much larger closed-source and open-source models on motion benchmarks.

## 5 ANALYSIS

The experimental results in the previous section demonstrate the high quality of our dataset; fine-tuning models with only *46.7k* videos (*467k* QAs) already leads to substantial improvements in motion understanding. In this section, we analyze the dataset, including ablation studies on the data curation process (Sec. 5.1) as well as the data distribution and overall statistics (Sec. 5.2).

### 5.1 DATA CURATION RELATED ANALYSIS

Our data curation pipeline rests on two key factors. (i) By leveraging object detection and trajectory tracking, we extract precise spatial relations and motion trajectories of all objects in the videos and feed them into LLMs to generate detailed captions and QA pairs. (ii) We design five complementary QA types that jointly capture diverse aspects of spatial relationships and motion understanding. In the following sections, we evaluate the contribution of each factor.

Table 2: Comparison of QA quality from video-only vs. video+bounding box JSONs, evaluated by GPT-4. Scores are normalized to 0–10 (higher is better) and averaged over three runs.

| Evaluation Dimension | Video Only | Video + BBox JSONs | Gain |
|---|---|---|---|
| Fine-grained Action Accuracy | 5.8 | **8.4** | +2.6 |
| Motion Detail and Specificity | 6.1 | **8.7** | +2.6 |
| Temporal Coherence | 6.5 | **8.9** | +2.4 |
| Question Relevance | 6.9 | **8.5** | +1.6 |
| Overall QA Quality | 6.3 | **8.6** | +2.3 |

**Bounding Box JSONs Improve Caption and QA Generation.** To assess the effect of structured object annotations, we compare QA generation with two input settings to LLMs: (i) raw video input and (ii) video with bounding box JSONs. We use GPT-4 as the evaluator (see prompts in Appendix A.4). As shown in Table 2, setting (ii) achieves higher scores across all dimensions, yielding substantial gains, particularly in fine-grained action accuracy (+2.6), motion detail and specificity (+2.6), and temporal coherence (+2.4). These improvements highlight the role of bounding boxes in providing structured spatial signals that help disambiguate subtle motions (e.g., hand reaching, object sliding) and support richer, temporally coherent QA generation. In contrast, video-only input often produces generic and less precise descriptions.

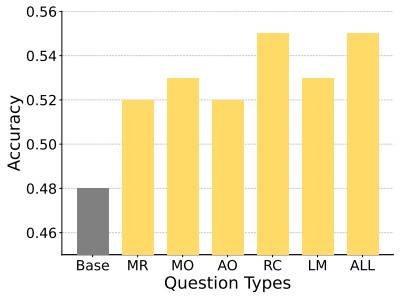

Figure 4: Impact of different question types on model accuracy.

**Different QA Pair Types Provide Complementary Benefits.** We have five different question types, and in this section we study their impact on model performance. We take Qwen2.5-7B as the base model and fine-tune it with 2,000 data samples for each experiment. As shown in Figure 4, every motion-focused question type outperforms the baseline (Base = 48%). Motion Recognition (MR) and Action Order (AO) each reach 52% (+8.3% over Base), Motion-related Objects (MO) and Location-related Motion (LM) both achieve 53% (+10.4%), and Repetition Count (RC) delivers the largest gain at approximately 55% (+14.6%).

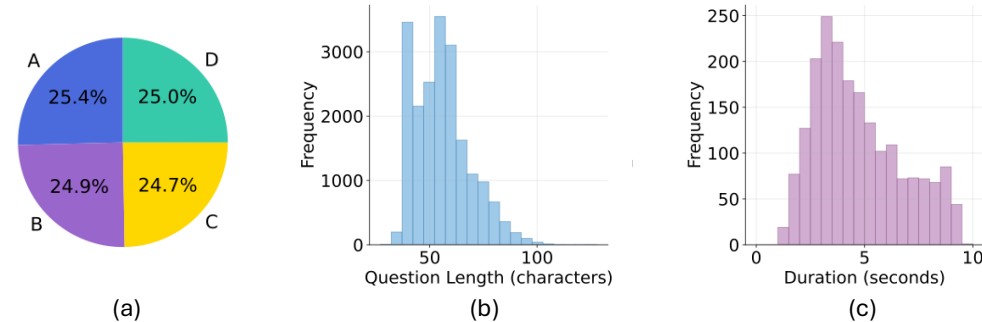

Figure 5: Dataset statistics. (a) correct answer distribution across options, (b) question length distribution measured in characters, and (c) video duration distribution in seconds.

The aggregated setting (ALL) also attains 55%, indicating that combining types matches the best single-type improvement and stabilizes performance. The ranking is $RC \approx ALL > MO \approx LM > MR \approx AO \gg Base$, suggesting that categories demanding explicit temporal integration and counting (RC) add the most, while object/spatial grounding (MO/LM) and coarse motion recognition/ordering (MR/AO) contribute complementary, mid-sized gains. Overall, the diverse QA types target distinct error modes—temporal precision, object–motion association, and spatial grounding—whose combined coverage yields consistent improvements over the baseline.

## 5.2 DATA DISTRIBUTION OF THE FOUNDATIONMOTION DATASET

The FoundationMotion Dataset consists of *46.7k* videos and *467k* QAs, where each QA pair consists of a question, four options, an answer, and a category. The task distribution is displayed in Fig. 5. Fig. 5(a) shows that the correct answers are evenly distributed across the four options, indicating no annotation bias. Fig. 5 (b) illustrates the distribution of question lengths measured in characters, where most questions fall between *30* and *80* characters. Fig. 5 (c) reports the distribution of video durations, which are mostly concentrated within *3–7* seconds, ensuring that the dataset emphasizes short but motion-rich clips. Together, these statistics highlight that FoundationMotion provides a balanced QA design, concise yet informative questions, and temporally compact videos well-suited for motion-centric video understanding.

We show the distributions of correct answers, question lengths, and video durations in Figure 5. The correct answers are nearly uniformly distributed across the four options (A–D), ensuring no bias toward a particular choice. Question lengths are concentrated between 40 and 70 characters, suggesting that the majority of questions are concise while still containing sufficient descriptive detail. Video durations primarily range from 2 to 6 seconds, providing short yet information-rich clips that balance annotation efficiency with motion diversity. Together, these statistics indicate that the dataset is well-balanced and designed to support reliable evaluation of video–language models.

## 6 CONCLUSION

In this paper, we propose FoundationMotion, an automated motion labeling pipeline for generalized spatial detection, tracking, and understanding of object behaviors. We demonstrate that fine-tuning with the FoundationMotion Dataset on various "how" motion benchmarks enables existing open-source VLMs to outperform larger models, and even compete with or surpass some closed-source models such as Gemini-2.5-Flash.

**Limitations and Future Work.** While FoundationMotion has achieved significantly strong results as demonstrated, its current spatial understanding is primarily limited to 2D space. Understanding "how" objects move in 3D remains a challenging but essential step toward a more comprehensive understanding of the real world. For example, while we demonstrate hand movement in this paper, understanding how each joint moves to form dexterous hand motions in 3D space would greatly benefit robotics and related applications. We will continue to explore this direction and promise to release all our code, data, and benchmarks to support further development in this field.

## REPRODUCIBILITY STATEMENT

We are committed to ensuring the reproducibility of our work. All code, datasets, and benchmarks associated with FoundationMotion will be publicly released upon publication. Details of the data generation pipeline are provided in Sec. 3, while the implementation details of fine-tuning are described in Sec. 4.1. Our goal is to facilitate research on motion understanding by providing transparent and accessible resources, thereby raising awareness of its importance, establishing a common standard for the field, and fostering community development. We will continue to maintain and improve the FoundationMotion codebase and dataset to support long-term reproducibility.

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

# A   APPENDIX

## A.1   BASIC STATISTICS OF FOUNDATIONMOTION DATASET

Table 3 summarizes the overall statistics of the FoundationMotion dataset. On average, each video lasts 17.51 seconds and is paired with about 10 questions. This corresponds to an annotation density of 1.671 questions per second, indicating a high level of temporal granularity in QA annotations. The average question length is 55.9 characters, showing that the questions are concise yet sufficiently descriptive. Together, these statistics highlight that FoundationMotion provides dense and informative annotations over relatively short video clips, making it well-suited for evaluating motion-level understanding in video-language models.

Table 3: Overall statistics of the FoundationMotion dataset.

| Metric | Value |
|---|---|
| Average video duration | 17.51 seconds |
| Average questions per video | 10.04 |
| Average annotation density | 1.671 questions/second |
| Average question length | 55.9 characters |

## A.2   PROMPTS USED FOR CAPTION GENERATION

---

**Background**

You are a detailed video caption generation tool focusing on object motion analysis and spatial relationships. You generate comprehensive captions for videos based on the video itself and the provided object motion information drawn on the video and in JSON.

**Motion label**

The motion information for the video in JSON format is as follows {`motion_info`}. It captures bounding box locations for various objects and their interactions in each frame of the video.

In the JSON format, it starts with object id, e.g., `"object_00"`, `"object_01"`, `"object_02"`, etc. Under each object id, there are `"bbox"`, `"object_type"` and `"interactions"` keys for the object. The `"bbox"` key contains a list of bounding boxes of the object in each frame throughout the video. The `"object_type"` key specifies the category of the object (e.g., `"person"`, `"cup"`, `"ball"`, `"car"`, etc.).

Under `"interactions"`, there are lists of other objects that this object is interacting with or spatially related to in each frame. The bounding boxes are in the format of `[left, top, right, bottom]` where the values are normalized to [0, 1] according to video width and height as in `[left/width, top/height, right/width, bottom/height]`. If the object is not detected in the frame, the bounding box value will be `None` in the list at the corresponding frame index. If objects are not interacting with any other objects in the frame, then `"interactions"` will be `None` at the corresponding frame index.

The detected bounding boxes are also drawn on each frame of the video: different object types with labels on top of colored bounding boxes for easy identification.

---

## A.3 PROMPTS USED FOR QA GENERATION

Background and task

You are provided with a video and a video caption that describes object motions and spatial relationships. Your task is to generate a list of concise questions and corresponding answers that evaluate a viewer's understanding of object motion analysis and spatial relationships.

Requirements

Question coverage should focus on five main categories:
**1. Motion Recognition Questions:** - **Action description**: What action is [object/person] performing? (e.g., raising hand, skiing, cooking, walking, etc.) - **Activity identification**: Describe the specific motion or gesture being performed - **Behavior characterization**: What type of movement pattern is observed?
**2. Action Order Questions:** - **Temporal sequence**: Which action happens first/second/last? - **Action timing**: What action occurs before/after [specific action]? - **Sequential events**: In what order do the actions unfold?
**3. Motion-related Object Questions:** - **Actor identification**: Which object/person performs [specific action]? - **Object-action association**: What does [object] do in the video? - **Agent-activity linking**: Who or what is responsible for [specific motion]?
**4. Location-related Motion Questions:** - **Spatial motion context**: Where does [action] take place in the scene? - **Position-based activity**: What motion happens in the [left/right/center/upper/lower] part of the scene? - **Spatial properties**: How does the location affect or relate to the motion?
**5. Repetition Count Questions:** - **Frequency counting**: How many times does [action] occur? - **Repetitive patterns**: How often is [motion] repeated? - **Cyclical behaviors**: What is the count of [repeated action]?
**6. Traditional Motion Analysis Questions:** - **Direction**: Which direction does [object] move? (left, right, up, down, diagonal directions) - **Distance**: How far does [object] move? (specific measurements, relative distances) - **Velocity**: How fast does [object] move? (speed characteristics, acceleration patterns) - **Trajectory**: What path does [object] follow? (straight, curved, circular, zigzag patterns)
**7. Spatial Relationship Questions:** - **Relative positions**: Where is [object A] positioned relative to [object B]? (left/right/up/down/front/back) - **Distance relationships**: How far apart are [object A] and [object B]? - **Positional changes**: How does the spatial relationship between [object A] and [object B] change?

Answer requirements

- Answers must be concise and directly address the question. - Include specific directional terms, distance measurements, and spatial descriptors when available. - Do not include extra explanations or thought processes in the answers.

Task

First, generate a list of questions and answers as below, with an empty line between each question and answer pair. Do not include any other texts in the output. Q1: ... A1: ...
Here are example questions and answers: Q1: What action is the person performing with their right hand? A1: The person is raising their right hand above their head.
Q2: Which action happens first in the video? A2: The person picks up the cup before stirring.
Q3: What object performs the cutting motion? A3: The knife performs the cutting motion on the vegetables.
Q4: Where in the scene does the stirring action take place? A4: The stirring action takes place in the upper-left area of the kitchen counter.
Then, for each question and answer, turn the single answer into 4 multiple choices with reasonable choices generated from the caption but distinctive from the correct answer. Please make sure each choice in the four choices is distinctive and do not have ambiguity with any other choice. Check the video content to make sure to never generate ambiguous multiple choices for the same question. Always put the correct answer at the first choice.

Output format

The output format: output a list of strings and each string contains a question and its corresponding multiple choices as below. The number of questions equal to the number of items

in the list. Each question must have 4 choices listed, after A, B, C, D. [ 'Q1: ... A: ... B: ... C: ... D: ...',
... ]
The correct answer is always at A. Do not include any other texts in the output. with an empty line between each question and answer pair.
Focus on generating questions that test understanding of: - Motion recognition and action identification (raising hand, cooking, walking, etc.) - Action temporal sequences and ordering - Object-action associations and actor identification - Location-based motion analysis and spatial context - Repetition counting and frequency analysis - Object movement directions (left, right, up, down, diagonal) - Movement distances and trajectories - Movement speeds and velocity patterns - Spatial positioning (left/right/up/down relationships) - Changes in spatial arrangements - Object proximity and distance relationships
Please generate your questions and answers accordingly, focusing on motion analysis and spatial relationships described in the caption.

## A.4 PROMPTS USED FOR EVALUATE QA QUALITY

You are an expert evaluator of video-based question–answer generation.
Given two sets of QAs for the same video (Set A: generated with video only; Set B: generated with video + bounding box JSONs), rate each set independently on a scale of 0–10 for the following dimensions:
1. Fine-grained action accuracy (does the QA capture detailed actions precisely?)
2. Motion detail and specificity (does it describe how objects move, not just that they move?)
3. Temporal coherence (are the actions ordered and consistent over time?)
4. Question relevance (are the QAs relevant and informative about the video?)

## A.5 QUESTION-ANSWER EXAMPLES

**QA type 1: Motion Recognition**

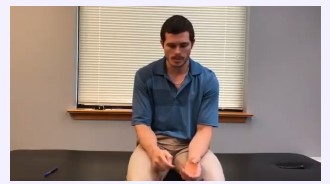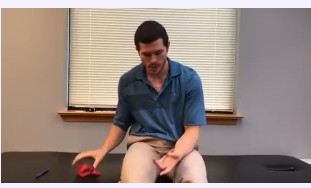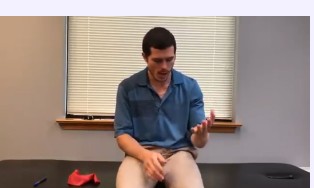

**What action is the person performing with their right hand?**

A. The person is raising their left hand.
B. The person is writing with a pen using their left hand.
C. **The person is manipulating the red object with their right hand.**
D. The person is resting both hands on their lap.

## QA type 2: Motion-related Objects

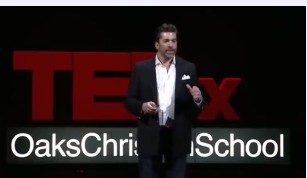 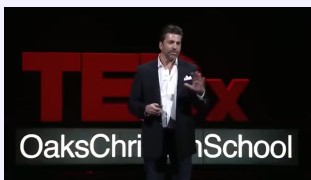 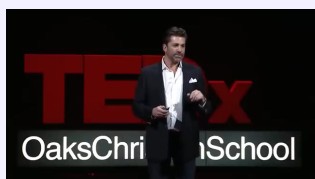

**What object performs the action of holding during the presentation?**

A. **The speaker's right hand holds an object, likely a microphone or remote.**

B. The speaker's right hand holds a glass of water.

C. The speaker's left hand holds a phone.

D. The speaker's left hand holds a notepad.

## QA type 3: Action Order

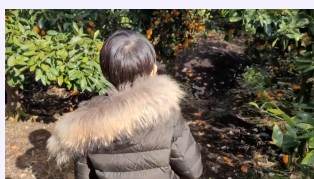 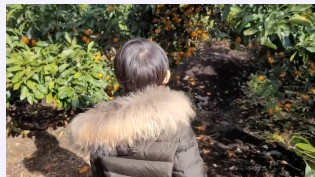 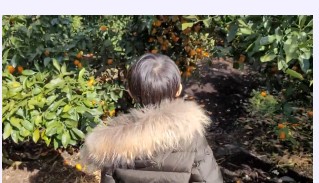

**Which action happens first in the video?**

A. The child picks an orange before standing still.

B. **The child stands still before reaching for the oranges.**

C. The child looks around before reaching for the oranges.

D. The child walks towards the oranges before reaching.

## QA type 4: Repetition Count

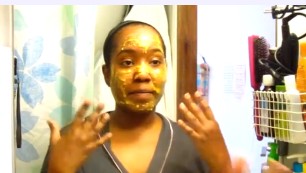 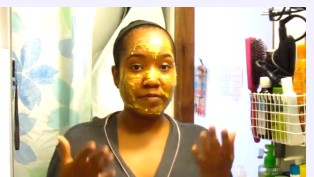 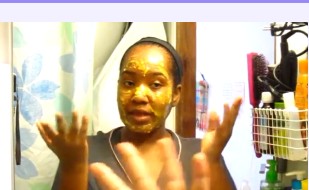

**5. How many times does the person gesture with their hands?**

A. The person gestures with their hands three times.

B. The person gestures with their hands only once.

C. The person does not gesture with their hands at all.

D. **The person gestures with their hands multiple times throughout the video.**

**QA type 5: Location-related Motion**

**Where in the scene does the walking action take place?**

A. **The walking action takes place along a path in the center of the frame.**

B. The walking action takes place on the left side of the frame.

C. The walking action takes place indoors.

D. The walking action takes place in a park.

