# OpenReview forum: "FoundationMotion: Auto-labeling and Reasoning about Spatial Movement in Videos"
_ICLR.cc/2026/Conference — Submitted to ICLR 2026_

### Official Review · Reviewer_uz9b · 2025-10-27

**Soundness:** 2
**Presentation:** 2
**Contribution:** 1
**Rating:** 2
**Confidence:** 5

**Summary:**

FoundationMotion is a new data pipeline designed to create large-scale motion datasets to address the scarcity of such data for training AI models. By detecting and tracking objects in videos, the pipeline generates detailed trajectories and uses large language models to create descriptive captions and question-answer pairs for physical reasoning.

**Strengths:**

- This paper targets a fundamental task of motion understanding in videos.

- The paper is well presented to show the results.

- The experimental results look good.

**Weaknesses:**

- The paper misses an important baseline, MotionLLM (chen et al. 2025), which is designed for motion understanding.

- The contributions listed by the authors overlap among these items. Why can the 4th point be the contribution? The 3rd point is just an experimental verification of the 1st one. Why can this be a contribution? The second contribution is not well described in the experiment part and lacks comparison with other benchmarks.

- The key contribution is not a new dataset, but relabelling an existing dataset. The authors should verify this and test the annotation pipeline in other video language datasets, like Valley or Video-ChatGPT.

- Besides, the scientific contribution is quite limited, but just some engineering ideas of designing labeling tools. These efforts are limited to bringing some novel insights to the community.

- The benchmark is not compared with existing ones.



My main concern lies in the overlapped and limited contributions. Besides, the experiments are not solid enough to verify the effectiveness of the pipeline.

**Questions:**

How do authors avoid the hallucination of LLMs?

The Grounding DINO is not very accurate. How can authors avoid the accumulated error?

How different scales of the relabelled data help the better result?

---

> ### Comment · Reviewer_uz9b · 2025-11-26
>
> As no response was provided, I keep my rating to reject this submission due to my listed concerns.

---

### Official Review · Reviewer_Tfgn · 2025-10-31

**Soundness:** 3
**Presentation:** 3
**Contribution:** 2
**Rating:** 4
**Confidence:** 4

**Summary:**

This paper introduces FoundationMotion, a pipeline for automatic data curation designed to generate large-scale motion understanding datasets for VLM. The proposed system integrates modern detection (SAM2, Grounded-DINO), tracking (SAM2), and reasoning (GPT-4o-mini) components to automatically extract object trajectories, generate detailed captions, and produce QA pairs focused on motion and spatial reasoning. The authors fine-tune several open-source models (NVILA-Video-15B/8B, Qwen2.5-VL-7B) using the curated data and evaluate them on both public and in-house benchmarks. Results show consistent gains in fine-grained motion reasoning tasks—sometimes outperforming much larger models e.g. Gemini-2.5-Flash, Qwen2.5-VL-72B.

**Strengths:**

Strong Engineering Contribution — The data curation pipeline is meticulously designed, combining open-vocabulary detection, hierarchical human-centric tracking, and structured LLM-based caption/QA generation.

High-Quality Evaluation — The authors not only evaluate on public motion reasoning benchmarks (MotionBench, VLM4D) but also construct four new domain-specific “how motion” benchmarks (Daily, Robotics, AV-Car, AV-Hand), enhancing reproducibility and completeness.

Clear Performance Gains — Consistent improvements across multiple models and tasks, with significant jumps on motion-intensive benchmarks (e.g., +17.8% on Robotics). The paper convincingly argues that FoundationMotion data is superior to other motion-focused datasets like PLM.

**Weaknesses:**

Limited novelty: The work is technically solid, but the major contribution is building a large, well-engineered pipeline rather than proposing new algorithms or models. It feels more like a strong system integration effort than a conceptual or methodological innovation. Moreover, automatic/semi-automatic data curation/labelling is a known technique.

Error propagation: Since the data is mainly curated automatically using existing models, errors from these models will propagate into the dataset.

The whole problem-solution also seem circular i.e. curating data by existing models to advance the capability of models. Moreover, it can cause evaluation bias as mentioned in the next point.

Possible evaluation bias: Since part of the evaluation is based on datasets that are generated with the same auto-labeling pipeline, there is potentially circular evaluation. Moreover, the improvements could simply be because additional data from the proposed pipeline was used and knowledge from so many other models was transferred.

Overemphasis on human-centric detection: The authors put a lot of design focus on detecting hands and distinguishing left vs. right hands, which is interesting, but they largely ignore other body parts. This feels overly tailored to human-hand motion and might not generalize well. It also relies heavily on human priors, which limits scalability and flexibility to other motion types.

Limited to 2D motion: The system only handles 2D trajectories, which makes sense but also caps its usefulness for robotics or 3D reasoning. Without depth or multi-view information, it’s hard to capture richer spatial dynamics.

Scalability and cost are unclear: The pipeline combines several heavy models—SAM2, LLMs, detection models—and it’s not clear how computationally expensive the whole process is. It would be good to see some runtime or resource analysis to understand whether this is practical for others to reproduce or scale up.

Some Minor Points:

Name of the pipeline and dataset: FoundationMotion is the name of the pipeline and the generated dataset. It can also be mistaken for a model.

Contribution statement: Contribution 4 is that “we will release our code, data and benchmarks”. Generally, this is not mentioned as a contribution.

Typo on page 5: “we also introduce” -> We also introduce

**Questions:**

Please refer to the Weaknesses.

---

### Official Review · Reviewer_dGJq · 2025-11-03

**Soundness:** 3
**Presentation:** 3
**Contribution:** 4
**Rating:** 4
**Confidence:** 3

**Summary:**

This paper addresses the gap in fine-grained motion understanding in VLMs, that a lack of how motion data hinders performance. The authors introduce FoundationMotion, a fully automated data curation pipeline. This pipeline detects and tracks objects in videos to extract 2D trajectories, then uses an LLM to generate high-quality, motion-focused captions and question-answer pairs based on these trajectories. The key finding is that fine-tuning open-source models on the resulting dataset (467k pairs) substantially boosts their motion reasoning, enabling them to outperform larger models like Gemini-2.5-Flash on specific motion benchmarks.

**Strengths:**

- Introduce a novel method that uses object trackers to create structured trajectory data (JSONs), which is then fed to an LLM to auto-generate a large-scale motion dataset.

- The paper provides a key ablation study showing that giving the model this structured tracking data results in much higher-quality questions and answers compared to giving it only the raw video frames.

**Weaknesses:**

- The method in paper filters out videos with significant camera motion. So what is the model's generalizable ability to real-world scenarios where the camera is moving?

- The paper lacks an analysis of error propagation. Since the pipeline is detect > track > caption, errors from upstream models (e.g., tracking failures) can be as noise or incorrect ground truth in the final dataset, yet the impact of this noise on model training is not measured.

- The method's reliance on 2D bounding boxes limits its ability to capture complex 3D interactions, such as the dexterous joint movements of a hand. So why 3D representations (like 3d bboxes) were not used?

**Questions:**

See in weakness. I would like to request a reconsideration of my score if these questions are addressed.

---

### Official Review · Reviewer_j17x · 2025-11-04

**Soundness:** 3
**Presentation:** 3
**Contribution:** 3
**Rating:** 4
**Confidence:** 4

**Summary:**

The authors propose an automated dataset pipeline to generate motion understanding data from videos. The pipeline includes object-level recognition and tracking, making it full of motion semantics in terms of different objects in the video. Besides, a "how" motion benchmark is proposed. With these training data, Qwen-2.5-VL-7B and NVILA-Video-8/15B have obvious performance gain and surpass Gemini-2.5-Flash and Qwen-2.5-VL-72B on several benchmarks, demonstrating the effectiveness of the training data.

**Strengths:**

1. The automated pipeline will contribute enough high-quality training data for model training in the community.
2. From the experiment results, these training data indeed provide an obvious performance boost for different models and even surpass the proprietary model Gemini-2.5-Flash.
3. A "how" motion benchmark is proposed. To some extent, it fills the gap of the lack of such motion questions (compared with the "what" questions).
4. In the Analysis section, ablation studies on the data generation process demonstrate the effectiveness of the pipeline design.
5. Overall, the paper is well-written.

**Weaknesses:**

Major:
1. In line 216, the function s_m = ..., what is the last term δ· max(∆r) representing?
2. In line 283, the authors mention the 7 dimensions of motion caption. Are there any examples or explanations? Or it is hard to understand, e.g., (7) the evolving spatial relationships.
3. What is the value of the proposed "how" benchmark? The authors only showcase the results in Table 1, but there is no in-depth analysis of how existing models have good performance on "what" questions, as mentioned in the introduction, and how they perform poor in "how" questions. For example, whether these models cannot clearly recognize the moving direction? Or the intention behind the motion?
4. The authors could probably better design specific question types for the benchmark, not just the video scenarios (car, hand, daily, etc.), so this could have a clear diagnosis of on what dimension these models have disadvantages.

Overall, the automated data pipeline is indeed a good contribution, but the lack of in-depth analysis of the model performance on "how" motion benchmark and the oversimplified design of the benchmark weaken its value.

Minor:
1. In line 53, the authors talk about the limitations of Gemini, but they did not provide solid evidence. Any examples? Since the car movement recognition can be very easy for current models.
2. In line 84, what is the logic flow between these 2 sentences, "Building a robust VLM ..." and "This means an annotator ..."? Looks like the writing needs improvement.

**Questions:**

Please refer to the weakness.

---

### Meta-Review · Area_Chair_XDhm · 2025-12-16

**Summary:**

This paper proposed an automated data annotation pipeline to generate motion understanding data for improving VLMs. Experimental results demonstrates the effectiveness of the data annotated by the proposed pipeline in improving VLMs. Besides, a "How" motion benchmark is also proposed to better evaluate the motion understanding of VLMs. These contributions are acknowledged by all reviewers.

Major concerns from the reviewers are:
1. Limited technical contributions (by Reviewer uz9b and Tfgn). The work is more towards a nice engineering work by gluing many existing techniques together, the technical contribution is limited.
2. Lack of in-depth analysis for some aspects (by Reviewer j17x, dGJq and Tfgnk). Reviewers have concerns about the in depth analysis about the "How" benchmark, the analysis about the error propagation and the bias introduced by the data pipeline as both training and benchmark data are labeled using the same data pipeline.
3. Extension to 3D motion. Both reviewer dGJq and Tfgn proposed concerns about extending the method to 3D motion understanding.

Overall, the reviewers's concerns are valid and I tend to reject this paper.

**Reviewer Concerns:**

As the authors did not get into the discussion. The reviewers's concerns and questions are not answered.

**Reviewer Scores:**

This paper gets score (4, 4, 4, 2), as the authors did not get into the discussion, I tend to believe the reviewers will keep their scores.

---

### Decision · Program_Chairs · 2026-01-26

Reject